# DRAG VIEW: GENERALIZABLE NOVEL VIEW SYNTHESIS WITH UNPOSED IMAGERY

## ABSTRACT

We introduce **DragView**, a novel and interactive framework for generating novel views of unseen scenes. DragView initializes the new view from a single source image, and the rendering is supported by a sparse set of unposed multi-view images, all seamlessly executed within a single feed-forward pass. Our approach begins with users *dragging* a source view through a local relative coordinate system. Pixel-aligned features are obtained by projecting the sampled 3D points along the target ray onto the source view. We then incorporate a view-dependent modulation layer to effectively handle occlusion during the projection. Additionally, we broaden the epipolar attention mechanism to encompass all source pixels, facilitating the aggregation of initialized coordinate-aligned point features from other unposed views. Finally, we employ another transformer to decode ray features into final pixel intensities. Crucially, our framework does not rely on either 2D prior models or the explicit estimation of camera poses. During testing, DragView showcases the capability to generalize to new scenes unseen during training, also utilizing only unposed support images, enabling the generation of photo-realistic new views characterized by flexible camera trajectories. In our experiments, we conduct a comprehensive comparison of the performance of DragView with recent scene representation networks operating under pose-free conditions, as well as with generalizable NeRFs subject to noisy test camera poses. DragView consistently demonstrates its superior performance in view synthesis quality, while also being more user-friendly. Codes will be released.

## 1 INTRODUCTION

Generalizable novel view synthesis, as demonstrated by recent works (Yu et al., 2021b; Wang et al., 2021a; Liu et al., 2022; Suhail et al., 2022a; Varma et al., 2022), has showcased the capability to generate new views on unseen scenes in a feed-forward manner. Despite their effectiveness, the prerequisite of rendering high-quality images, such as accurate multi-view posed images, has inhibited their practical usage. Prior research has predominantly relied on pre-computed poses (Schönberger & Frahm, 2016), or online-computed poses with multi-view geometry (Tian et al., 2022), learned prior knowledge (Smith et al., 2023). Nevertheless, the process requires a dense sequence as input, and the pose estimation can be time-consuming, and the incorrectly estimated pose may yield suboptimal rendering results. One could bypass the demand for camera poses by adopting only a single image to learn generalizable NeRFs (e.g., PixelNeRF (Yu et al., 2021b)), and render the target image from the constructed feature volume. On the other hand, Scene Representation Transformer (SRT) (Sajjadi et al., 2022) and RUST (Sajjadi et al., 2023) have expanded the ability to process multiple images as a "set latent scene representation" and generate novel views even in the presence of flawed camera poses or without any pose information. However, the problem of scene reconstruction under a single input is highly ill-posed and faces challenges on in-the-wild scenes, while the latent representation results in blurred rendering outcomes with a lower resolution (e.g., $128 \times 128$ for SRT), limiting their applicability in achieving photorealistic rendering.

In this work, we tackle the problem of novel view synthesis with unposed images from a fresh perspective. We propose to leverage pixel-aligned features for each target pixel, and incorporate image-based rendering (IBR) to warp and composite source images into the target view in a learnable manner. Nonetheless, the first key challenge lies in determining the location and orientation of the target view from a set of unposed images. Previous works (Sajjadi et al., 2022; 2023) rely on the

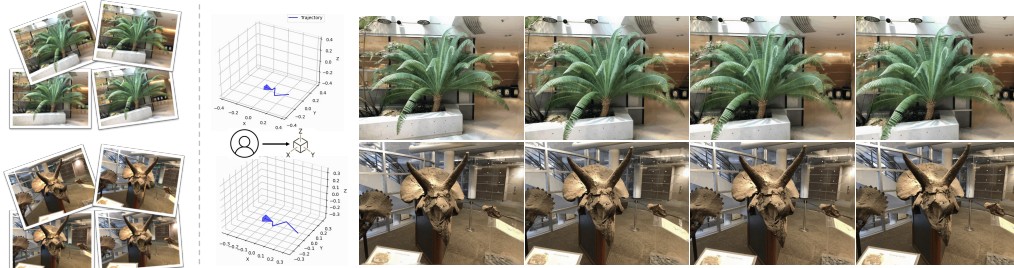

Unposed images (new scene) | User-controlled path | Novel view synthesis in a forward-pass, without per-scene optimization.

Figure 1: **Inference pipeline of DragView.** Given unposed multi-view images of an unseen scene, and the user-selected views to render, our approach generates novel views corresponding to a continuous long camera trajectory in a feed-forward pass, without per-scene optimization.

relative pose between one of the source views or partly preview the target view. We contend that selecting a source view as the initial viewpoint and enabling users to interactively explore the target view is more realistic. Consequently, users initially start from one of the source images and drag it through the mouse or the VR headset to specify the target view for rendering. This allows the generation of the target image in a single end-to-end forward pass, utilizing the unposed images.

Another challenge pertains to parameterizing the target view feature for predicting its final pixel intensities. Our solution involves the initialization of the pixel-aligned 3D features from the target view onto the source view by establishing a local relative transformation between the user-controlled target view and a user-selected source view. A view-conditioned modulation layer is introduced to mitigate the occlusion in the projection while preserving global coherence. A tokenization module and the designed OmniView Transformer that are used for the aggregation of the point-wise features from other unposed source views. The final pixel intensities along a target ray are decoded using another transformer.

By combining all these efforts, our approach — DragView, achieves superior rendering quality at higher resolutions compared to previous methods (see Figure 1). When compared to pose-free PixelNeRF and SRT series, our approach achieves the state-of-the-art performance on both synthetic objects and real-world scenes, without relying on assumptions about the availability of camera poses in unseen scenes. Additionally, our method demonstrates superior robustness against camera pose noise when compared to other generalizable NeRFs.

Our key contributions are:

- We introduce a new formulation for generalizable novel view synthesis from a sparse set of unposed images, eliminating the need for pose annotations among source views, and streamlining the rendering process.

- We establish a local relative coordinate system that is user-controlled between the new view and the selected source view. Point features are initialized through projection, and we propose source-conditioned modulation layers and cross-view attention mechanisms to manage both occlusions and the aggregation of the coordinate-aligned point features.

- In comparative evaluations, DragView consistently outperforms other pose-free, generalizable neural rendering techniques. Furthermore, it demonstrates remarkable robustness in managing variations in pose accuracy, leading to superior rendering quality compared to generalizable NeRF approaches.

## 2 RELATED WORKS

**Generalizable Neural Scene Representations**   Building generalizable feature volumes dates back to Neural Volumes (Lombardi et al., 2019), wherein an encoder-decoder framework is adopted to create a feature volume. Later on, NeRF (Mildenhall et al., 2021) and its follow-ups (Barron et al., 2021; 2022; Yu et al., 2021a; Fridovich-Keil et al., 2022; Liu et al., 2020; Hu et al., 2022; Sun et al., 2022; Müller et al., 2022; Wang et al., 2022; Garbin et al., 2021; Hedman et al., 2021; Reiser et al., 2021; Attal et al., 2022; Sitzmann et al., 2021; Feng & Varshney, 2021) have emerged as ef-

fective scene representations. However, their costly per-scene fitting nature constitutes a significant limitation. Generalizable NeRFs endeavor to circumvent time-consuming optimization by conceptualizing novel view synthesis as a cross-view image-based interpolation problem. NeuRay (Liu et al., 2022), IBRNet (Wang et al., 2021a), MVSNeRF (Chen et al., 2021), and PixelNeRF (Yu et al., 2021b) assemble a generalizable 3D representation using features aggregated from observed views. GPNR (Suhail et al., 2022a) and GNT (Varma et al., 2022) enhance the novel view renderings with a Transformer-based aggregation process. A view transformer aggregates image features along epipolar lines, while a ray transformer combines coordinate-wise point features along rays through the attention mechanism.

**Pose-free NeRFs** Numerous efforts have been exerted to diminish the necessity for calibrated camera poses during NeRF training. NeRF- - (Wang et al., 2021b) makes an early endeavor to optimize camera parameters with NeRF training for forward-facing scenes simultaneously. BARF (Lin et al., 2021) refines NeRFs from imperfect (or even unknown) camera poses via coarse-to-fine registration, while GARF (Chng et al., 2022) incorporates Gaussian activations. NoPe-NeRF (Bian et al., 2023) employs monocular depth priors to restrict the relative poses between consecutive frames. Efforts have also been made to expand generalizable NeRF toward unposed images. PixelNeRF (Yu et al., 2021b) builds a generalizable feature volume that estimates novel views from single-view observation, which can be unposed. SRT (Sajjadi et al., 2022) and RUST (Sajjadi et al., 2023) infer a set-latent scene representation from a set of unposed images to synthesize novel views. MonoNeRF (Tian et al., 2022), when provided with a video sequence, explicitly estimates the depth maps, camera poses, along with a reconstruction loss between the rendered frames for training. FlowCam (Smith et al., 2023) explicitly estimates the camera pose of the video frame by fitting a 3D scene flow field with the assistance of a pretrained 2D optical flows model.

**Transformers as Neural Scene Representations** Transformers are extensively utilized to represent scenes. IBRNet (Wang et al., 2021a) processes points sampled from rays using MLP and estimates density via a transformer. NeRFormer (Reizenstein et al., 2021) employs attention modules to craft feature volumes with epipolar geometry constraints. LFNR (Suhail et al., 2022b) introduces a two-stage transformer-based model to amass features along epipolar lines and aggregate features along reference views to produce the color of target rays. SRT (Sajjadi et al., 2022) and RUST (Sajjadi et al., 2023) infer a set-latent scene representation via a vision transformer and parameterize light fields by attending into the scene representation for novel view renderings. GPNR (Suhail et al., 2022a) and GNT (Varma et al., 2022) initially utilize view transformers to accumulate multiview features along the epipolar lines on source views and then adopt a ray transformer to decode the features for novel views.

## 3 METHOD

**Overview** Our method begins with an unordered set of *unposed images* with known intrinsics $\{(\boldsymbol{I}_i \in \mathbb{R}^{H \times W \times 3}, \boldsymbol{K}_i \in \mathbb{R}^{3 \times 3})\}_{i=0}^N$ of a new scene. The objective is to synthesize new views within this scene in a single forward pass. Users select the "starting" view $\boldsymbol{I}_0$ as the origin. The target view, determined by the user's mouse or headset, establishes the relative coordinate system. The target ray is emitted, and 3D points are sampled along it. The initial coordinate-aligned 3D point features are determined by projecting onto the starting view, with missing or occluded regions compensated by employing a view-dependent modulation layer. A specially designed *OmniView Transformer* is utilized to aggregate features from unposed images towards point-wise features, and the pixel intensities are subsequently decoded. To determine the most suitable source views for aggregation, a view selector is designed to identify the closest $K$ views ($K \leq N$) based on global feature distances. An overview of our pipeline is illustrated in Figure 2.

### 3.1 INTERACTIVE TARGET FEATURE INITIALIZATION

**Tokenizing Coordinate-aligned Points** To capture the multi-scale 2D features of the input image set $\{\boldsymbol{I}_i \in \mathbb{R}^{H \times W \times 3}\}_{i=0}^K$, where $K$ denotes the number of source views used for generating the target image, a shared 2D CNN (He et al., 2016) is employed across all source views to construct hierarchical image features. For each target pixel, coordinate-aligned 3D points, denoted as $\{p_1, \cdots, p_M\}$, are uniformly sampled within the range determined by the near and far planes. These points are subsequently projected onto the feature maps extracted from the starting view, culminating in the

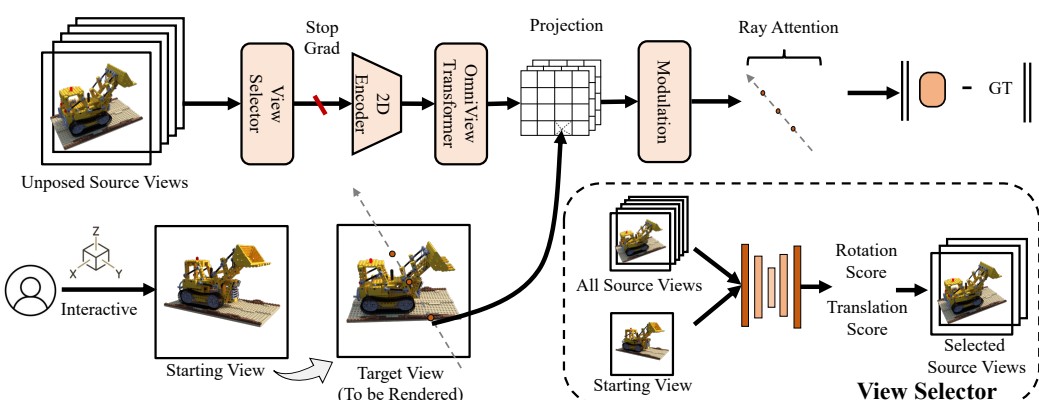

Figure 2: **The overall pipeline of the proposed DragView.** Given unposed source images and a user-selected starting view, DragView selects a limited number of source images closest to the starting view. Multi-scale 2D features are extracted, and the 3D points on the target ray are projected. View-dependent modulation layers are introduced to address occlusion, followed by the OmniView Transformer to aggregate features from unposed source images to the target 3D points. The pixel intensities are then decoded via another Transformer.

initialized point features:

$$\hat{p_i}^0 = \boldsymbol{K}^0(\boldsymbol{R}^0(p_i) + t^0) \tag{1}$$

$$\mathcal{F}(\boldsymbol{F}^0, \hat{p_i}^0) = \text{Interpolation}(\Pi(\hat{p_i}^0, \boldsymbol{F}^0)) \tag{2}$$

Here, $\boldsymbol{R}^0$ and $t^0$ symbolize the relative rotation and translation between the target view to the starting view, $\Pi$ represents the projection function, and $\boldsymbol{F}^0$ is the extracted feature maps on starting view, formulating the pixel-aligned point-wise feature $\{\boldsymbol{f}_p^1, ..., \boldsymbol{f}_p^M\}$.

**View-Conditioned Modulation Layer**  When projecting points into regions of the scene that are obscured from the camera's viewpoint, the projected points often become occluded or missed, diminishing the rendered quality. To counteract this obstacle, a dedicated conditional layer is introduced to modulate the point features through an affine transformation, utilizing the global coherence derived from the starting view. Formally, a Feature-wise Linear Modulation (FiLM) layer (Perez et al., 2018) is utilized to scale and shift the point feature $\boldsymbol{f}_p^i$ following:

$$\gamma_i^\tau, \beta_i^\tau = \text{MLP}_\gamma(\text{GAP}(\boldsymbol{F}^0)), \text{MLP}_\beta(\text{GAP}(\boldsymbol{F}^0)) \tag{3}$$

$$\boldsymbol{f}_m^i = \gamma_i^\tau \boldsymbol{f}_p^i + \beta_i^\tau \tag{4}$$

This modulation is orchestrated by two group parameters, $\gamma_i^\tau$ and $\beta_i^\tau$, resulting in the modulated point-wise feature $\{\boldsymbol{f}_m^1, ..., \boldsymbol{f}_m^M\}$. GAP denotes the global average pooling. A prominent advantage of utilizing FiLM layers lies in their ability to alleviate the issues associated with unsatisfactory projections. They achieve this by leveraging the comprehensive feature adaptation derived from the entire starting view, while incurring minimal computational overhead, attributed to the employment of a single-layer multi-layer perceptron (MLP).

## 3.2 OMNIVIEW TRANSFORMER

In the domain of multi-view stereo and novel view synthesis, the understanding of camera poses facilitates the utilization of the *Epipolar Transformer* (He et al., 2020; Yang et al., 2022; Suhail et al., 2022a). This transformer functions between the pixels in the target view and the pixels positioned on the epipolar line of the posed source views. Such a global receptive field is beneficial to aggregate long-range context information within and across images. However, it is essential to understand that explicitly constructing epipolar geometry is unfeasible when working with unposed image sets. Moreover, in many circumstances, the rendered pixel intensity emanates from a blend of both direct light—which adheres to the epipolar geometry moving from the source to the camera, typically reflecting at most once within the scene—and indirect light—that follows non-epipolar paths, involving multiple reflections, especially in scenes with intricate light transport (O'Toole et al., 2014).

Given these elements, we propose the utilization of the "attention" mechanism as a universal trainable aggregation function, to enrich the pixel-aligned features from all source views, culminating in the design of the *OmniView Transformer*.

**Elevating Epipolar Attention to All Pixels**    The OmniView architecture comprises self-attention and cross-attention layers, each with multiple heads. The formal representation of the attention mechanism is defined as $\mathrm{Attention}(\boldsymbol{Q}, \boldsymbol{K}, \boldsymbol{V}) = \mathrm{softmax}\left(\frac{\boldsymbol{Q}\boldsymbol{K}^T}{\sqrt{C}}\right)\boldsymbol{V}$, where $\boldsymbol{Q}$, $\boldsymbol{K}$, and $\boldsymbol{V}$, located within $\mathbb{R}^{N \times C}$, denote the query, key, and value matrices, respectively, where $N$ represents the flattened token count, and $C$ indicates the hidden dimension. The most direct method for aggregating the modulated feature $\{\boldsymbol{f}_m^1, ..., \boldsymbol{f}_m^M\}$ from all pixels in other source views proves intractable as the attention complexity will be quadratic to the token number. Hence, we suggest an efficient strategy by: **1).** tokenizing each view by subdividing the extracted feature map $\boldsymbol{F}$ into $M \times M$ patch grids, introducing self-attention and cross-attention layers for both intra-view and inter-view aggregations toward the "starting view". In our approach, the self-attention layers first independently process each source view, utilizing the view itself as both the query ($\boldsymbol{Q}$) and key ($\boldsymbol{K}$), effectively enabling intra-image long-range global context aggregation. Subsequently, we amalgamate features from all other source views $\{\boldsymbol{I}_i \in \mathbb{R}^{H \times W \times 3}\}_{i=1}^K$ towards the "starting view ($\boldsymbol{I}_0$)," which acts as the root of the constructed local coordinate system. For this purpose, the "starting view", is assigned as key ($\boldsymbol{K}$), while all other source views function as query ($\boldsymbol{Q}$). **2).** projecting the $\{p_1, \cdots, p_M\}$ to attain the initialized pixel-aligned feature $\{\boldsymbol{f}_p^1, ..., \boldsymbol{f}_p^M\}$, on the aggregated feature planes. The pseudocode implementation of OmniView Transformer is provided in Appendix.

**Ray-based Rendering with Transformers**    NeRF (Mildenhall et al., 2021) achieves photorealistic rendering by integrating the color and density along the ray cast from the target pixel, a pivotal element in NeRF's success. Nevertheless, it remains a simplified form of volume rendering, based on specific physical assumptions (Max, 1995). Recent research advocates the utilization of the Transformer architecture to adaptively learn the blending weights along the ray for each point, augmenting both expressiveness (Varma et al., 2022) and generalization (Suhail et al., 2022a). In alignment with these advancements, we employ such an "attention" mechanism to determine the aggregation weights for each sampled point feature, further decoding into pixel intensities in a learnable fashion.

### 3.3    View Selection from Global Feature Matching

View selection aims to select efficiently a few source images, which is the nearest to the starting view, to reduce the computational redundancy when performing OmniView attention. Specifically, a network is designed to extract multi-scale features (He et al., 2016) from all source images, and multiple decoding heads are devised for regressing the relative rotation and translation scores between $\mathbf{I}_0$ and each source image $\{\mathbf{I}_i, i \neq 0\}$. In particular, four decoding heads are utilized for estimating the three normalized relative angles and the distance value between the two images. Top $K$ images are selected out of the $N$ ($K \leq N$).

### 3.4    Training and Inference

During the training phase, the view selector identifies the nearest $K$ source images from the $N$ unposed source images. This selection procedure is guided by a specified loss function that operates based on the online-computed relative angle and distance values of each image pair.

$$\Theta_s^* = \arg\min_{\boldsymbol{\Theta}}(\|\angle(\boldsymbol{I}_0, \boldsymbol{I}_i) - \angle_{gt}\|_2^2 + \|d(\boldsymbol{I}_0, \boldsymbol{I}_i) - d_{gt}\|_2^2). \tag{5}$$

The remainder of the model is optimized utilizing the $\mathcal{L}_2$ distance between the rendered target pixels and the corresponding ground-truth pixels, as exemplified by:

$$\Theta_t^* = \arg\min_{\boldsymbol{\Theta}} \|\boldsymbol{C}(\boldsymbol{r}_i | \boldsymbol{\Theta}, \boldsymbol{\theta}, \boldsymbol{x}) - \boldsymbol{C}_{gt}\|_2^2. \tag{6}$$

For evaluation, there is no reliance on any pose estimator to render a new viewpoint. Instead, target views are specified based on a user-selected starting view. The ground-truth target pose is subsequently transformed into a relative one for computing quantitative metrics. For the free-view rendered video, both the starting view and the other source views are held constant.

Table 1: **Quantitative Comparison in a Generalizable Pose-Free Setting.** DragView is assessed alongside several generalizable methods on both synthetic and real-world datasets. The adopted "starting view" serves as a reference frame for both PixelNeRF and UpSRT, ensuring a fair evaluation. DragView outperforms previous pose-free methods utilizing both single-view feature volume (Yu et al., 2021b) and multi-view "set of latent" (Sajjadi et al., 2022). The best-performing method is highlighted in bold.

| Methods | Real Forward-facing(LLFF) | | | | NeRF Synthetic Objects | | | | RealEstate10K Datasets | | | |
|---|---|---|---|---|---|---|---|---|---|---|---|---|
| | PSNR↑ | SSIM↑ | LPIPS↓ | Avg.↓ | PSNR↑ | SSIM↑ | LPIPS↓ | Avg.↓ | PSNR↑ | SSIM↑ | LPIPS↓ | Avg.↓ |
| PixelNeRF | 8.379 | 0.313 | 0.643 | 0.539 | 7.105 | 0.565 | 0.413 | 0.422 | 9.008 | 0.407 | 0.503 | 0.466 |
| UpSRT | 14.154 | 0.467 | 0.592 | 0.455 | 11.421 | 0.783 | 0.306 | 0.281 | 13.483 | 0.525 | 0.464 | 0.399 |
| Ours | **22.728** | **0.778** | **0.180** | **0.219** | **22.832** | **0.835** | **0.134** | **0.181** | **24.208** | **0.789** | **0.190** | **0.217** |

## 4 EXPERIMENTS

### 4.1 IMPLEMENTATION DETAILS

**Training Datasets.** The training data encompasses both synthetic and real datasets collected by IBRNet (Wang et al., 2021a). The synthetic subset contains 1,023 models from the Google Scanned Objects (Downs et al., 2022). Real datasets comprise sparsely sampled scenes from RealEstate10K (Zhou et al., 2018), 100 scenes from the Spaces datasets (Flynn et al., 2019), and 102 real-world scenes captured using handheld phones (Mildenhall et al., 2019; Wang et al., 2021a). Ground truth poses are utilized during training to optimize the view selector. Importantly, the inference pipeline does not utilize the camera poses during rendering.

**Evaluation Datasets.** The widely adopted synthetic data created by (Mildenhall et al., 2021) encompasses eight objects with a resolution of 400×400. Additionally, eight real-world forward-facing scenes from the LLFF datasets (Mildenhall et al., 2019) are included, each at a resolution of 504 ×378. Four scenes from the test set in RealEstate10K (Zhou et al., 2018) of 176×144 resolution are also employed for computing quantitative metrics and generating visualizations.

**Training and Inference Details.** DragView is trained end-to-end, with the gradient stopping operation after the view selector. The Adam optimizer is employed to minimize the training loss for the model. The learning rate decreases exponentially over training steps with a base of $10^{-3}$. The comprehensive training encompasses 250,000 steps, with 4,096 randomly sampled rays during each iteration. In both training and inference phases, 128 points are uniformly sampled along each target ray. Grid number $M$ is set as 7 to balance efficiency and accuracy. We implement the UpSRT (Sajjadi et al., 2022) based on their provided model details, as they do not publish the code, and UpSRT is trained on the same datasets with us.

**Baselines and Error Metrics.** We evaluate DragView against two categories of generalizable methods: *pose-free neural rendering* on general scenes and *generalizable NeRFs* employing multi-view noisy posed images. In terms of the pose-free method, we make comparisons with the state-of-the-art method UpSRT (Sajjadi et al., 2022). UpSRT defines the target view in relation to one source view, aligned with our approach. We also compare with single-view based PixelNeRF (Yu et al., 2021b), which utilizes a reference image to construct the feature volume, enabling rendering on the target views through projection and volume rendering. Regarding the generalizable neural rendering employing posed images (Wang et al., 2021a; Varma et al., 2022; Liu et al., 2022), we synthetically perturb the camera poses with additive Gaussian noise on both Rotation and translation vectors similar to BARF (Lin et al., 2021). The added noises are intended to simulate different degrees of erroneous pose computation, and to test the robustness of existing methods against the camera noises. We set the standard deviation from 0.003 to 0.1 on source (support) views for all tested methods. The rendering quality is reported using metrics such as peak signal-to-noise ratio (PSNR), structural similarity (SSIM), and perceptual similarity via LPIPS (Zhang et al., 2018).

### 4.2 POSE-FREE NOVEL VIEW SYNTHESIS ON UNSEEN SCENES.

Table 1 showcases that DragView surpasses the best-performing multi-view pose-free SRT and single-view PixelNeRF on all unseen test data, encompassing synthetic datasets, unbounded forward-facing datasets, and RealEstate10K datasets. This validates the efficacy of the designed interactive framework. DragView not only leverages the information within the starting view but

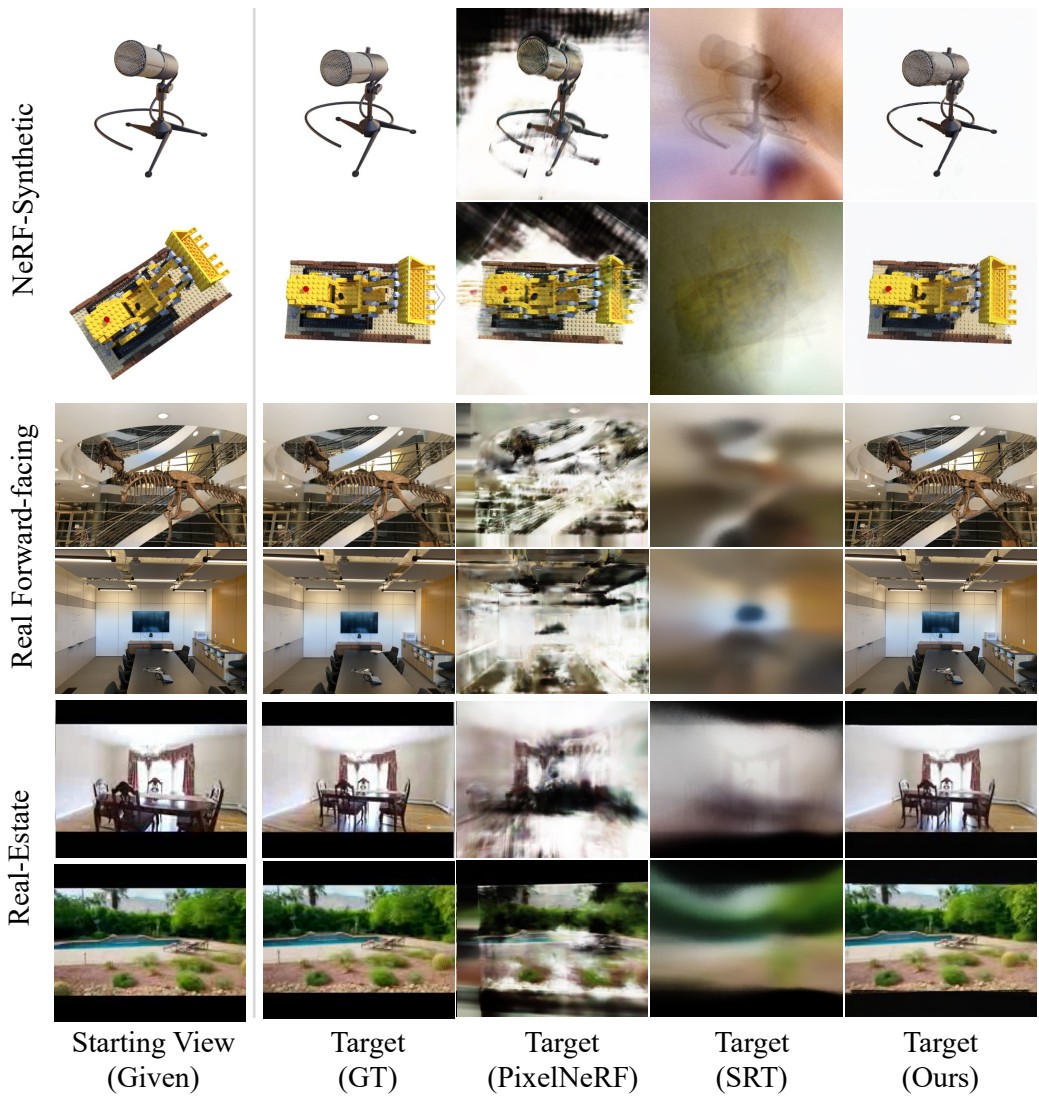

Figure 3: **Qualitative comparison under generalizable pose-free setting.** Our method can more accurately recover fine details, and produce images that are significantly more similar to the ground truth. Single-view PixelNeRF (Yu et al., 2021b) introduces artifacts with the constructed feature volume, and shows insufficient capacity on complex scenes. Whereas multi-view SRT (Sajjadi et al., 2022) fails to render sharp details for the scenes with rich textures. Note that all methods utilize the "Starting View" as the reference frame for the relative coordinate system.

also effectively aggregates pixel-aligned features from other unposed source views, as it can fill the unseen regions when move to a new view. We present qualitative results in Figure 3, wherein it can be observed that the "latent" representation in SRT overlooks the image details, and Pixel-NeRF struggles under complex scenes using single-view feature volume-based neural rendering. See the video in supplementary materials for more detailed comparisons.

### 4.3 ROBUSTNESS AGAINST NOISY POSES.

Multi-view images captured in the real world typically necessitate a pre-processing step (e.g., COLMAP (Schönberger & Frahm, 2016)) to compute the poses. However, the computation is slow, especially when the number of source images is large, and often contains errors (Lin et al., 2021). We examine the current best-performing generalizable NeRFs against noisy camera poses in the tested

Table 2: **Quantitative comparison of robustness to noisy poses in source views.** The table presents a performance comparison between DragView and various generalizable NeRFs using the NeRF-Synthetic and LLFF datasets, where both rotation and translation matrices are perturbed with $\sigma = 0.003$. Notably, the best-performing methods are indicated in bold, while the second-best performers are underlined. DragView showcases its robustness in handling pose perturbations in rendered views.(remove view selector, use GT poses for selecting K nearest views, add noise)

| Methods | Real Forward-facing(LLFF) | | | | NeRF Synthetic | | | |
|---------|-------------|------|--------|-------|-------------|------|--------|-------|
| | PSNR↑ | SSIM↑ | LPIPS↓ | Avg.↓ | PSNR↑ | SSIM↑ | LPIPS↑ | Avg.↓ |
| IBRNet | 21.395 | 0.686 | 0.303 | 0.290 | 20.027 | 0.813 | 0.145 | 0.195 |
| NeuRay | 21.520 | 0.681 | 0.303 | 0.291 | 21.424 | 0.832 | 0.135 | 0.184 |
| GNT | 21.341 | 0.682 | 0.307 | 0.292 | 20.554 | 0.830 | 0.139 | 0.186 |
| Ours | **22.728** | **0.778** | **0.180** | **0.219** | **22.832** | **0.835** | **0.134** | **0.181** |

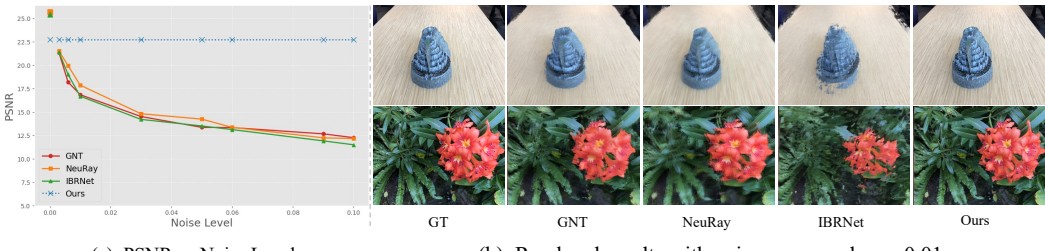

(a). PSNR vs Noise Levels          (b). Rendered results with noisy poses under σ=0.01.

Figure 4: Visualizations of different methods against noisy poses on source images when rendering. All adopted generalizable NeRFs suffer from noisy camera poses in source views at evaluation, even with very mild perturbation (e.g., $\sigma$=0.003). When the noise increases, methods that rely on camera poses for cross-view aggregation are decreasing in rendering quality. Our method demonstrates the robustness against test pose noises.

source views, a practical concern. Following (Lin et al., 2021), who apply additive Gaussian perturbation at different levels to the camera poses, we directly test the trained generalizable model with the provided checkpoint to assess robustness. It is clear from Figure 4 (left) that all generalizable methods suffer from noisy camera poses, with a significant degradation in performance even under a small amount of noisy calibration ($\sigma$=0.003). We can also observe that stronger noises continuously degrade the rendering metrics. DragView, which generates new views in a feed-forward pass without estimating camera poses, demonstrates stability in the rendering quality. Figure 4 (right) visualizes the effect of different noise levels on source views in the evaluation, where we can see subtle noise significantly decreases the rendering quality. Quantitative results with noise level as 0.003 on both real forward-facing and synthetic objects datasets are shown in Table 2.

## 4.4 ABLATION STUDY

We now execute a series of ablation studies regarding our module choice on the LLFF datasets (Mildenhall et al., 2019) and average the metrics across all scenes. The evaluation begins with the use of the "starting view" only, and subsequently, we incrementally integrate the proposed techniques in this study.

**Pixel-aligned Feature?** We employ the interactive framework to establish the relative coordinate system between the "starting" and "target" views. In the target view, pixel-aligned features are initialized by projecting onto the starting view. The final pixel intensities are regressed using ray attention, without feature propagation from other views. The numbers in Table 3 validate that exploring pixel-aligned features surpasses the variant that uses the "starting view" for evaluation, but still lags behind our full model.

**OmniView Transformer?** We further employ the *OmniView Transformer* to aggregate multi-view patch features using a data-driven attention mechanism toward the starting view. This approach notably enhances quantitative metrics, with the averaged error decreased from 0.396 to 0.237, a 40% improvement, as is shown in the third row of Table 3. Besides, Figure 5 demonstrates the visual quality is also improved by integrating missing information from other source views.

**Conditional Modulation Layer?** Conditioned on the statistics of the starting view, the pixel-aligned features are affine-transformed with learnable parameters. This adjustment aids in complet-

Table 3: **Ablation study of the proposed components in DragView.** We report the metrics for evaluating the effectiveness for each proposed module.

| Starting View | Pixel-aligned Feature | OmniView Transformer | Modulation FiLM-Layer | PSNR↑ | SSIM↑ | LPIPS↓ | Avg.↓ |
|:---:|:---:|:---:|:---:|:---:|:---:|:---:|:---:|
| ✓ | ✗ | ✗ | ✗ | 14.198 | 0.355 | 0.407 | 0.415 |
| ✗ | ✓ | ✗ | ✗ | 16.519 | 0.412 | 0.400 | 0.396 |
| ✗ | ✓ | ✓ | ✗ | 22.287 | 0.740 | 0.197 | 0.237 |
| ✗ | ✓ | ✓ | ✓ | **22.728** | **0.778** | **0.180** | **0.219** |

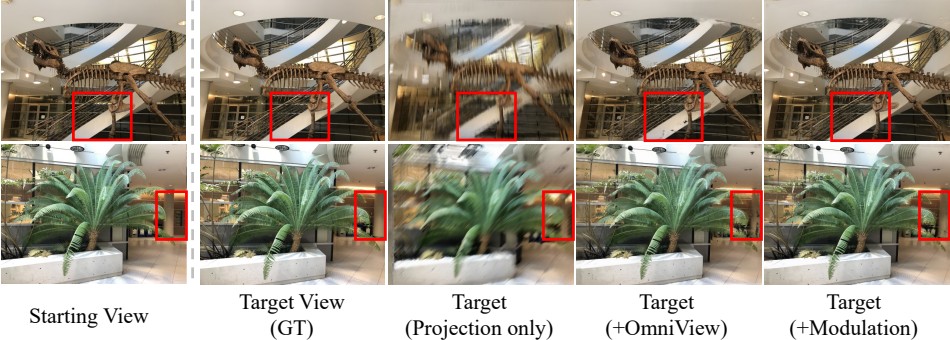

| Starting View | Target View (GT) | Target (Projection only) | Target (+OmniView) | Target (+Modulation) |

Figure 5: **Visualization for Ablation Study.** We visualize the starting view", ground-truth target view", rendered target with projection and ray attention, and rendered target that gradually incorporates the *OmniView Transformer* and the *Modulation Layer*.

ing the missing regions in occluded areas (refer to Figure 5) when the initial projection is imprecise which may lead to future erroneous aggregation. The last row of Table 3 shows such modulation enhances the averaged quantitative metrics by 7%.

**Analysis on the Viewpoint Selector.** Initially, we assess the use of random source view selection in the evaluation, where the selected source views may not be ideal for feature aggregation, resulting in a 0.505 SSIM metric (see Table 4). Utilizing the selector to regress the averaged relative viewing direction and distance directly elevates the SSIM to 0.756, a significant improvement. Employing separate decoder heads to disentangle the similarity score for the three axes of relative rotation and translation distance further enhances the quantitative metrics to 0.778 (↑ 0.02 in SSIM). Additionally, we illustrate the use of ground-truth poses to identify the nearest source views with viewing directions most analogous to the "starting view," serving as the upper bound of the view selector.

Table 4: **Ablation on the View Selector.** The incorporation of the view selector boosts the reduction of averaged error by a large margin, enabling DragView to effectively aggregate cross-view features. The employment of disentangled decoder heads for each axis of rotation and translation vector further improves the view selection accuracy and, thus, the synthesis quality. DragView, which utilizes ground-truth poses for view selection, is listed in the last row.

| Random?/GT?/ View Selector? | Disentangled Decoder Heads | PSNR↑ | SSIM↑ | LPIPS↓ | Avg.↓ | Select Acc.↑ |
|:---:|:---:|:---:|:---:|:---:|:---:|:---:|
| Random | N/A | 17.841 | 0.505 | 0.390 | 0.3700 | 0.017 |
| View Selector | ✗ | 22.243 | 0.756 | 0.198 | 0.2326 | 0.688 |
| View Selector | ✓ | 22.728 | 0.778 | 0.180 | 0.2188 | 0.731 |
| GT | N/A | 24.275 | 0.822 | 0.135 | 0.1869 | 1.000 |

## 5 CONCLUSION

We present an interactive framework, DragView, for photo-realistic rendering from a sparse set of unposed images. DragView utilizes pixel-aligned features and an efficient OmniView Transformer to aggregate unposed source views using a data-driven attention mechanism. DragView enjoys the advantages of Image-Based Rendering (IBR) and efficient cross-view aggregation to produce state-of-the-art rendering quality on complex scenes. DragView can render new views on unseen datasets without any scene-specific optimization and pre-computed camera poses, showcasing both the flexibility for pose annotations and robustness against noisy computed camera poses.

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

## A  EPIPOLAR TRANSFORMER VS. OMNIVIEW TRANSFORMER

As has been mentioned in the main paper, knowing the multi-view camera poses enables the framework design to search correspondences along epipolar line. As is shown in Figure 6(a), we consider a simple case when the source view number is two. For the pixel to be rendered in the target view, epipolar attention used in (Varma et al., 2022; Suhail et al., 2022a) build correspondences among the target ray and epipolar line of neighboring source images. However, without knowing the poses, we fail to build such a search pattern, and thereby, we utilize an attention mechanism to search over all source pixels toward the starting view, which is the origin of the relative coordinate system. We propose to utilize a CNN network to extract multi-scale feature maps. Subsequent to the CNN encoder, these extracted feature maps from source views $\{(\boldsymbol{I}_i \in \mathbb{R}^{H \times W \times 3})\}_{i=0}^{K}$ are subdivided into $M \times M$ grids within each view, enabling the model agonistic to diverse image resolutions (as is shown in Figure 6 (b)). The intra-view attention, which performs intra-image long-range global context aggregation, can retrieve relevant information within the given view (Figure 6 (c)), while the inter-view attention is performed separately to capture cross-relationships across these two views and inter-image feature interaction between images is done in this way (Figure 6 (d)). Detailed video explanations can be found in the supplementary material.

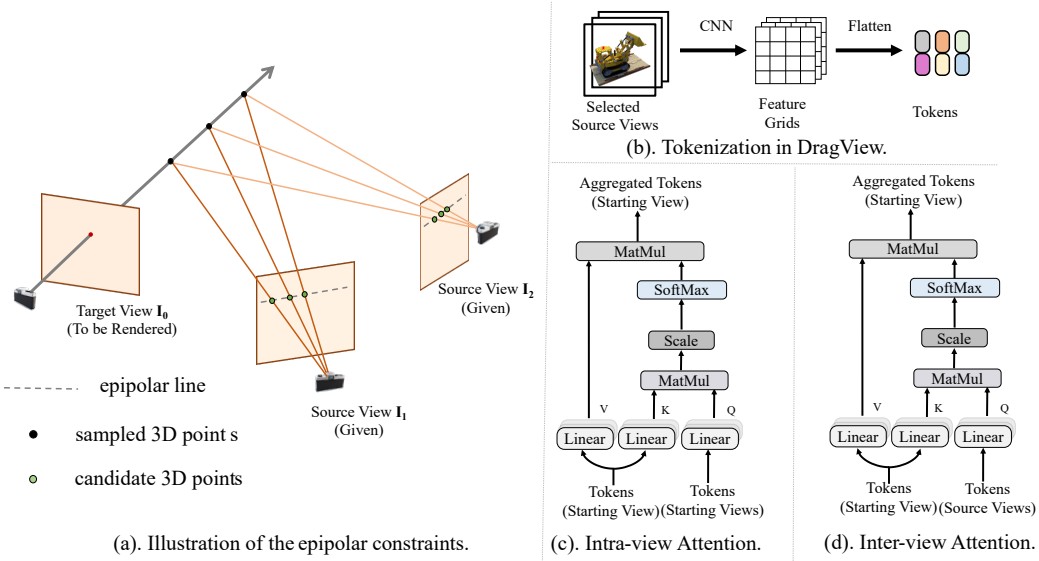

Figure 6: Illustration of *Epipolar Attention* and *OmniView Attention*.

## B  IMPLEMENTATION DETAILS

**Memory-Efficient OmniView Transformer**   The most straightforward method for aggregating the initially projected 3D point feature involves building cross-attention between the target 3D point feature and all source pixels. However, this approach is intractable as it cannot scale to high-resolution input images and a large number of source views. Therefore, we propose to leverage the $8\times$ downsampled CNN features and pool them into a fixed number of 2D grids (here, we use a $7\times7$ grid) for each view. Consequently, our design is agnostic to input resolution, allowing attention to be performed in a patch-wise manner. Nevertheless, during training, the sampled ray is typically large (e.g., 4096 in DragView), incurring $4096 \times 128$ sampled points in each iteration. The cross-attention among sampled points and tokenized patches in the source views remains intractable. Therefore, akin to the [CLS] token in Vision Transformer (Dosovitskiy et al., 2020), we employ the cross-attention mechanism to propagate multi-view information in source views toward the starting view. We then project the sampled 3D points onto the starting view, ensuring an efficient implementation regardless of the number of source views used. Please refer to the PyTorch-like pseudo-code Algorithm 1 for a detailed explanation.

---

**Algorithm 1** OmniView Transformer: PyTorch-like Pseudocode

---

$\boldsymbol{p}_t \to$ points coordinate in target view$(N_{\text{rays}}, N_{\text{pts}}, 3)$
$\boldsymbol{X}_0 \to$ flattened patch tokens in starting view$(1, N_{\text{patch}}, C)$
$\{\boldsymbol{X}_i\}_{i=1}^K \to$ flattened patch tokens in selected source views$(K, N_{\text{patch}}, C)$
$\boldsymbol{f}_t \to$ projected points feature along target ray$(N_{\text{rays}}, N_{\text{pts}}, D)$
$f_Q, f_K, f_V, f_{\text{rgb}} \to$ functions that parameterize MLP layers

**for** $0 \le i \le K$ **do**                                                    ▷ Intra-view attention
    $\boldsymbol{Q} = f_Q(\boldsymbol{X_i})$
    $\boldsymbol{K} = f_K(\boldsymbol{X_i})$
    $\boldsymbol{V} = f_V(\boldsymbol{X_i})$
    $\boldsymbol{A} = \text{matmul}(\boldsymbol{Q}, \boldsymbol{K}^T)/\sqrt{D}$
    $\boldsymbol{A} = \text{softmax}(\boldsymbol{A}, \dim = -1)$
    $\boldsymbol{X_i} = \text{matmul}(\boldsymbol{A}, \boldsymbol{V})$
**end for**

**for** $1 \le i \le K$ **do**                                                    ▷ Inter-view attention
    $\boldsymbol{Q} = f_Q(\boldsymbol{X_i})$
    $\boldsymbol{K} = f_K(\boldsymbol{X_0})$
    $\boldsymbol{V} = f_V(\boldsymbol{X_0})$
    $\boldsymbol{A} = \text{matmul}(\boldsymbol{Q}, \boldsymbol{K}^T)/\sqrt{D}$
    $\boldsymbol{A} = \text{softmax}(\boldsymbol{A}, \dim = -1)$
    $\boldsymbol{X_0} = \text{matmul}(\boldsymbol{A}, \boldsymbol{V})$
**end for**

**for** $0 \le i \le (N_{\text{rays}} \times N_{\text{pts}})$ **do**                              ▷ Point-wise projection
    $\boldsymbol{f}_t^i = \text{interp.}(\text{proj.}(\text{modulation}(\boldsymbol{p}_t^i), \boldsymbol{X}_0))$
**end for**

**for** $0 \le i \le N_{\text{rays}}$ **do**                                              ▷ Ray attention
    $\boldsymbol{Q} = f_Q(\boldsymbol{f}_t^i)$
    $\boldsymbol{K} = f_K(\boldsymbol{f}_t^i)$
    $\boldsymbol{V} = f_V(\boldsymbol{f}_t^i)$
    $\boldsymbol{A} = \text{matmul}(\boldsymbol{Q}, \boldsymbol{K}^T)/\sqrt{D}$
    $\boldsymbol{A} = \text{softmax}(\boldsymbol{A}, \dim = -1)$
    $\boldsymbol{f}_t^i = \text{matmul}(\boldsymbol{A}, \boldsymbol{V})$
**end for**

$\text{RGB} = f_{\text{rgb}}(\text{mean}_{i=1}^{N_{\text{pts}}}(\boldsymbol{f}_t^i))$

---

Table 5: Comparison of DragView with other pose-free generalizable novel view-synthesis methods on the NeRF Synthetic Datasets (scene-wise).

| Models | Chair | Drums | Ficus | Hotdog | Materials | Mic | Ship | Lego |
|--------|-------|-------|-------|--------|-----------|-----|------|------|
| PixelNeRF | 7.202 | 7.747 | 7.426 | 6.925 | 7.092 | 7.026 | 6.312 | 7.113 |
| SRT | 14.299 | 11.845 | 14.272 | 10.627 | 9.541 | 13.193 | 7.362 | 10.231 |
| DragView | 25.104 | 19.192 | 21.785 | 22.712 | 27.359 | 25.140 | 16.533 | 21.019 |

(a) PSNR↑

| Models | Chair | Drums | Ficus | Hotdog | Materials | Mic | Ship | Lego |
|--------|-------|-------|-------|--------|-----------|-----|------|------|
| PixelNeRF | 0.604 | 0.574 | 0.628 | 0.603 | 0.570 | 0.619 | 0.401 | 0.523 |
| SRT | 0.846 | 0.800 | 0.858 | 0.804 | 0.767 | 0.875 | 0.578 | 0.738 |
| DragView | 0.871 | 0.835 | 0.822 | 0.875 | 0.800 | 0.881 | 0.677 | 0.817 |

(b) SSIM↑

| Models | Chair | Drums | Ficus | Hotdog | Materials | Mic | Ship | Lego |
|--------|-------|-------|-------|--------|-----------|-----|------|------|
| PixelNeRF | 0.375 | 0.414 | 0.351 | 0.418 | 0.416 | 0.372 | 0.529 | 0.432 |
| SRT | 0.228 | 0.291 | 0.177 | 0.349 | 0.333 | 0.196 | 0.519 | 0.357 |
| DragView | 0.083 | 0.154 | 0.098 | 0.087 | 0.174 | 0.046 | 0.402 | 0.126 |

(c) LPIPS↓

Table 6: Comparison of DragView with other pose-free generalizable novel view-synthesis methods on the forward-facing LLFF datasets (scene-wise).

| Models | trex | fern | flower | leaves | room | fortress | horns | orchids |
|--------|------|------|--------|--------|------|----------|-------|---------|
| PixelNeRF | 8.266 | 8.655 | 8.234 | 7.026 | 8.872 | 10.550 | 7.743 | 7.177 |
| SRT | 13.999 | 13.848 | 14.396 | 12.676 | 15.669 | 16.597 | 12.945 | 12.306 |
| DragView | 21.489 | 21.847 | 22.786 | 17.725 | 26.836 | 27.261 | 23.866 | 16.139 |

(a) PSNR↑

| Models | trex | fern | flower | leaves | room | fortress | horns | orchids |
|--------|------|------|--------|--------|------|----------|-------|---------|
| PixelNeRF | 0.351 | 0.326 | 0.240 | 0.127 | 0.492 | 0.418 | 0.275 | 0.161 |
| SRT | 0.505 | 0.469 | 0.439 | 0.230 | 0.714 | 0.518 | 0.448 | 0.265 |
| DragView | 0.798 | 0.737 | 0.773 | 0.674 | 0.848 | 0.820 | 0.804 | 0.590 |

(b) SSIM↑

| Models | trex | fern | flower | leaves | room | fortress | horns | orchids |
|--------|------|------|--------|--------|------|----------|-------|---------|
| PixelNeRF | 0.618 | 0.645 | 0.658 | 0.668 | 0.603 | 0.582 | 0.669 | 0.738 |
| SRT | 0.551 | 0.600 | 0.655 | 0.710 | 0.453 | 0.554 | 0.620 | 0.722 |
| DragView | 0.181 | 0.208 | 0.158 | 0.285 | 0.133 | 0.136 | 0.171 | 0.312 |

(c) LPIPS↓

## C  ADDITIONAL EXPERIMENTS

**Scene-wise Quantitative Metrics**    Table 5, Table 6 and Table 7 include a scene-wise quantitative results presented in the main paper. Our method quantitatively surpasses both the generalizable single-view based method PixelNeRF (Yu et al., 2021b) and multi-view based method SRT (Sajjadi et al., 2022). We also include videos to demonstrate our results in the attached video.

Table 7: Comparison of DragView with other pose-free generalizable novel view-synthesis methods on the real Real-Estate datasets (scene-wise).

| Models | 0bcef | 000db | 000eb | 8516c |
|---|---|---|---|---|
| PixelNeRF | 8.541 | 9.284 | 10.084 | 8.055 |
| SRT | 13.348 | 13.410 | 14.121 | 13.055 |
| DragView | 24.760 | 22.808 | 23.487 | 25.778 |

(a) PSNR↑

| Models | 0bcef | 000db | 000eb | 8516c |
|---|---|---|---|---|
| PixelNeRF | 0.427 | 0.380 | 0.401 | 0.373 |
| SRT | 0.522 | 0.519 | 0.544 | 0.513 |
| DragView | 0.804 | 0.750 | 0.785 | 0.816 |

(b) SSIM↑

| Models | 0bcef | 000db | 000eb | 8516c |
|---|---|---|---|---|
| PixelNeRF | 0.507 | 0.515 | 0.486 | 0.504 |
| SRT | 0.461 | 0.463 | 0.458 | 0.474 |
| DragView | 0.174 | 0.220 | 0.193 | 0.172 |

(c) LPIPS↓

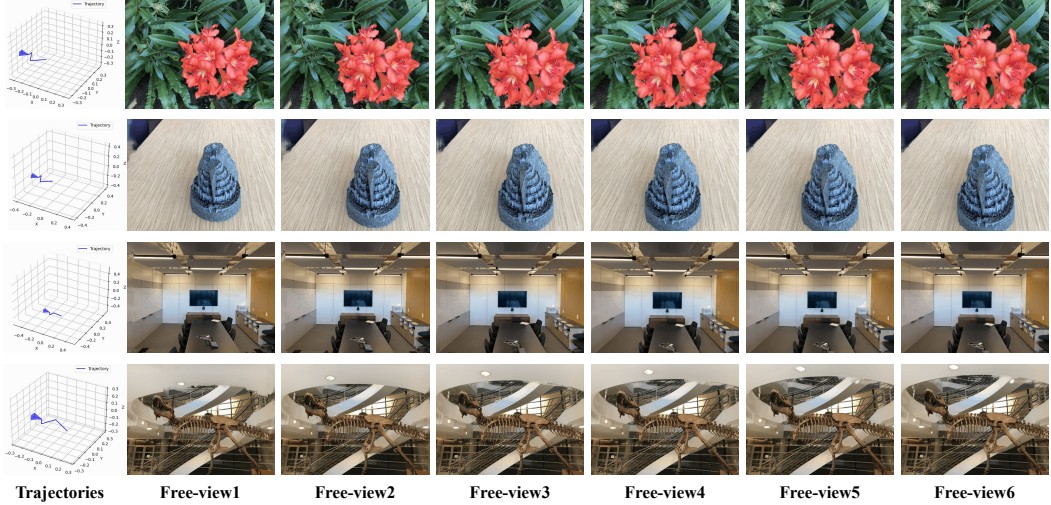

| Trajectories | Free-view1 | Free-view2 | Free-view3 | Free-view4 | Free-view5 | Free-view6 |
|---|---|---|---|---|---|---|

Figure 7: Visualization on more interpolated viewpoints. The visualized images showcase the efficacy of our method in handling various viewpoints by interpolating between test views on real-world datasets. See the video in supplementary materials for more detailed comparisons.

**More Free-View Rendering** We provide the visualization of multiple rendered novel viewpoints for the real-world dataset by interpolating between test views. This visualization, illustrated in Figure 7, demonstrates the capabilities of our method in generating diverse viewpoints, offering insight into its performance and potential limitations in real-world scenarios. Our approach consistently yields visually consistent new views when the dragged new view contains moderate unseen regions from the starting view, as previously discussed.

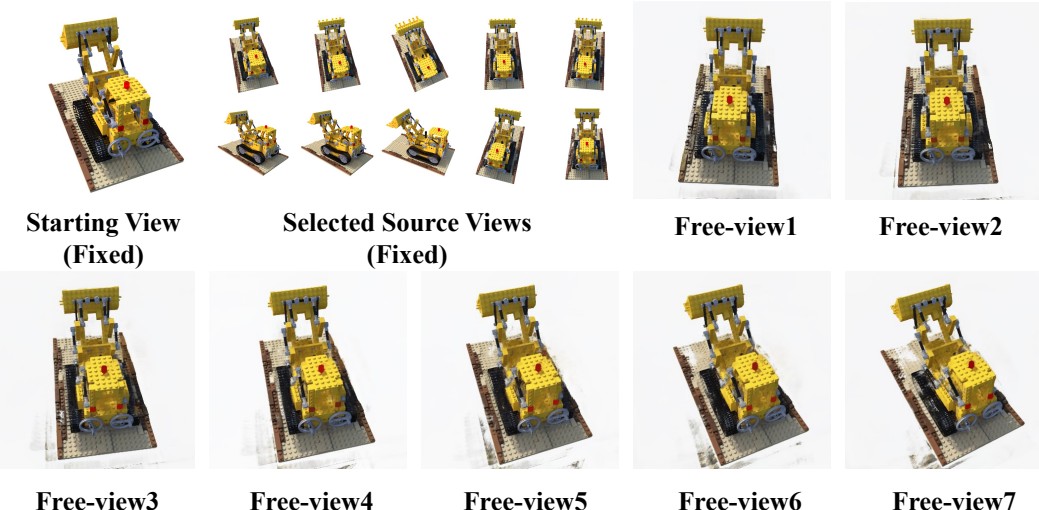

| Starting View (Fixed) | Selected Source Views (Fixed) | Free-view1 | Free-view2 |
|---|---|---|---|

| Free-view3 | Free-view4 | Free-view5 | Free-view6 | Free-view7 |
|---|---|---|---|---|

Figure 8: When the rendered target view encompasses substantial unseen areas from the starting view, the rendering quality decreases due to the imprecise initialization of these point features.

## D    LIMITATIONS

Our approach carries certain limitations inherited from these posed methods (e.g., IBRNet (Wang et al., 2021a)). For instance, while our method yields visually consistent new views when the dragged new view contains moderate unseen regions from the starting view, it does not generalize effectively to new views with significantly occluded regions (see Figure 8). This limitation arises because the initialized point-aligned feature cannot maintain accuracy in the presence of extensive occlusion, leading to a potential failure in subsequent aggregation based on these imprecise initializations. In Figure 8, as we fix the starting view and gradually move the target view to another side of the object, the rendering results begin to deteriorate. Updating the starting view by incorporating the newly generated views may mitigate this question by enhancing the accuracy and precision of the initialized point-aligned features, thus potentially improving the rendering quality even in the presence of the selection of extreme new views.

