# OpenReview forum: "Drag View: Generalizable Novel View Synthesis with Unposed Imagery"
_ICLR.cc/2024/Conference — ICLR 2024 Conference Withdrawn Submission_

### Official Review · Reviewer_iswE · 2023-10-29

**Soundness:** 2 fair
**Presentation:** 1 poor
**Contribution:** 2 fair
**Rating:** 3
**Confidence:** 4

**Summary:**

The paper proposes a model that, at test time, ingests a set of unposed image observations of a scene as input, and then enables novel view synthesis of the underlying 3D scene. No test-time optimization is necessary, this proceeds in a single feed-forward pass.

Notably, while the authors call their method "pose-free", they *do* require ground-truth poses at training time.

Overall, I had trouble parsing this paper and understanding the exact method (in particular the OmniVision Transformer), even after reading the paper several times and looking at the appendix. I'm further very well versed in this literature and thus assume that my fellow reviewers will face similar challenges.

Further, *if* I understand correctly what this paper is doing, it is somewhat incremental: it is essentially pixelNeRF / Generalized Patch-Based Neural Rendering with source views where their pose is unknown, but the pose of the target view *is* known. To compensate for the unknown source poses, the authors propose to use a multi-view transformer encoder. These concepts, however, are not novel, and - in combination with the lacking exposition - the paper does not pass my bar for publication.

I thus believe that this paper would need a significant amount of work on the exposition to make it clear, to a degree where it would require another round of reviewing.

**Strengths:**

- The problem setting - novel view synthesis from a set of unposed images - is well-motivated.
- Relatively comprehensive related work.

**Weaknesses:**

- The exposition of the paper is unfortunately unclear to a degree that I am not sure what it exactly is that the authors are proposing:
1. The input-output behavior of the method is confusing. Really, the paper simply assumes a set of input images among which the relative pose is unknown, as well as a target pose whose pose relative to *one* of the input images is known. This setting is identical to that of the unposed Scene Representation Transformer. This input-output behavior should be stated clearly in the introduction and the methods section. Instead, the authors "muddy the waters" by describing *how* the target pose is obtained relative to one of the input views, via an interactive "dragging" by the user. This is completely irrelevant to the method and should simply be removed. Figure 1 should be updated accordingly - the "user-contolled path" is simply irrelevant to the exposition of the method.
2. The method should be put into the context of prior work much more clearly. In essence, the method is like a version of pixelNeRF or GPNR where *several* input images are supplied whose relative pose is unknown, and the authors propose a particular attention-based mechanism to aggregate information across views in absence of known relative poses. This explanation would be significantly more clear than discussing "user inputs".
3. Figure 2 does not provide an overview over the actually critical part of the architecture, which is the "OmniVision Transformer". The OmniVision transformer is, unfortunately, unclear to me. Specifically, I do not understand how each of the source views acts as a "query" to perform cross-attention over the starting view "key", but then generates a feature map that is aligned with the starting view coordinate system? It would make much more sense to me if each token in the starting view would perform cross-attention over all the other source views, generating one feature map per additional source view, and we then use these additional feature maps in pixel-aligned rendering. But that is not what appears to be happening...?

- The authors should clarify that their method requires ground-truth poses at training time, and that, in particular, they assume that at training time, the pose of the target view relative to one of the input views is known. This is in contrast to methods such as FlowCam, which do *not* assume known camera poses even at training time.
- I have significant doubts about the performance of the UpSRT baseline. The results on NeRF-synthetic are not representative: if the UpSRT was trained on a large enough dataset and with appropriate settings and compute, one would expect at least a blurry reconstruction of the target object - in contrast, on the NeRF-synthetic dataset, the method simply does not perform at all. There is likely a bug or the model was not trained at a large enough scale.

**Questions:**

Please see weaknesses.

---

### Official Review · Reviewer_EhPc · 2023-11-01

**Soundness:** 2 fair
**Presentation:** 2 fair
**Contribution:** 2 fair
**Rating:** 5
**Confidence:** 4

**Summary:**

This paper presents a novel method for synthesizing new views in a scene using a sparse set of unposed images. The proposed approach utilizes an OmniView Transformer to gather features from the unposed images and employs image-based rendering (IBR) techniques to warp and composite source images into the target view. Notably, this method eliminates the requirement for pose annotations among the source views and streamlines the rendering process, thereby enabling generalizable novel view synthesis.

**Strengths:**

1. The proposed approach, DragView, introduces a novel formulation for generalizable novel view synthesis from unposed images, eliminating the need for pose annotations.
2. DragView demonstrates superior rendering quality at higher resolutions compared to previous methods.

**Weaknesses:**

1. The evaluation of DragView against noisy camera poses is limited to a small range of noise levels.
2. It could be better to discuss potential limitations or failure cases of DragView.

**Questions:**

1. What are the limitations of DragView, if any, in terms of scene complexity or dataset requirements?
2. How does DragView handle noisy camera poses in source views? Could you provide quantitative results on its robustness against different levels of pose noise?
3. Could you provide a comparison of DragView with other state-of-the-art generalizable NeRF approaches, like IBRNet, and RegNeRF, on both synthetic objects and real-world scenes?

---

### Official Review · Reviewer_cpwu · 2023-11-01

**Soundness:** 2 fair
**Presentation:** 2 fair
**Contribution:** 2 fair
**Rating:** 3
**Confidence:** 4

**Summary:**

The paper presents "DragView," an innovative framework designed for generating novel views of unseen scenes. DragView initializes new views from a single source image, supported by a sparse set of unposed multi-view images, all within a single feed-forward pass. A distinctive feature of DragView is its interactive nature, allowing users to drag a source view through a local relative coordinate system to determine the desired new view. The framework employs advanced techniques such as pixel-aligned feature projection, a view-dependent modulation layer to handle occlusions, and a broadened epipolar attention mechanism to aggregate features from other unposed views. Experimental results show that it outperforms existing methods.

**Strengths:**

The authors extended the existing view synthesis technology to a commercial level by connecting the network with a user interface, which might be possible to service customers not limited to academia.

**Weaknesses:**

- User-controlled interface may not be a novel contribution to the machine learning society but it is over-emphasized.
- I doubt that the existing methods in the experimental results are sufficiently trained, as the output images are over-corrupted.
- It is hard to find technical novelty as employing existing modules is insufficient, and methods for unposed images are typical.
- The epipolar attention mechanisms are also existing work named as Epipolar Transformer.

**Questions:**

No questions